# Nested helicoids in biological microstructures

Israel Greenfeld [1,2]*, Israel Kellersztein [1,2] & H. Daniel Wagner [1]*

Helicoidal formations often appear in natural microstructures such as bones and arthropods exoskeletons. Named Bouligands after their discoverer, these structures are angle-ply laminates that assemble from laminae of chitin or collagen fibers embedded in a proteinaceous matrix. High resolution electron microscope images of cross-sections through scorpion claws are presented here, uncovering structural features that are different than so-far assumed. These include in-plane twisting of laminae around their corners rather than through their centers, and a second orthogonal rotation angle which gradually tilts the laminae out-of-plane. The resulting Bouligand laminate unit (BLU) is highly warped, such that neighboring BLUs are intricately intertwined, tightly nested and mechanically interlocked. Using classical laminate analysis extended to laminae tilting, it is shown that tilting significantly enhances the laminate flexural stiffness and strength, and may improve toughness by diverting crack propagation. These observations may be extended to diverse biological species and potentially applied to synthetic structures.

---

[1] Department of Materials and Interfaces, Weizmann Institute of Science, Rehovot 76100, Israel. [2]These authors contributed equally: Israel Greenfeld, Israel Kellersztein *email: green_is@netvision.net.il; Daniel.Wagner@weizmann.ac.il

Helicoidal microstructures are often observed in biology, appearing in the exoskeleton of arthropods and in bones. First described by the French biologist Y. Bouligand in 1965[1,2] and named after him, a Bouligand structure consists of stacks of layers (laminae) of unidirectional chitin or collagen fibers embedded in a proteinaceous matrix, helically placed on top of each other in a way resembling plywood[3–5]. The intricate conformation of the Bouligand laminate unit (BLU) has captured the interest of scientists and engineers, particularly because of its resemblance to laminated composites and its potential for inspiring even stronger, stiffer and tougher synthetic structures[6–15]. The potential mechanical properties of BLUs have been analyzed by laminate theory[11,12] and fracture mechanics[12,16–19], and by building and testing macroscale models[19–23]. Reproducing BLUs at the microscale, the scale typically found in nature, is still an open ongoing challenge. Even more so is the assembly of many BLUs to form large tightly packed structures, and the investigation of their function and benefits.

The current literature describes the BLU as a laminate possessing infinite helical symmetry, with stacked laminae that are progressively twisted in-plane around an axis located at the lamina center (see for example[12–14,19,24]). Mathematically, this means that at any point along the twisting rotation axis, the helicoid will appear exactly the same[25]. However, our present study on the cuticle of the scorpion chela (pincers), using the *Scorpio Maurus Palmatus* as a model animal, reveals a more complex structural arrangement, which might be found in other species as well. In this arrangement the twisting axis is located at the lamina corner instead of its center, and in addition the laminae are progressively tilted by an out-of-plane rotation, resulting in a warped helicoid. This helicoid is neither infinite nor does it possess helical symmetry, as its appearance varies along the rotation axis. We use throughout the text the attribute 'asymmetric' to describe this type of helicoid.

High resolution SEM (scanning electron microscope) and TEM (transmission electron microscope) images of the cuticle of the scorpion tarsus (moveable claw), focusing on the endocuticle layer containing the BLU arrays, reveal the details of these helicoidal constructs, and show how they are nested and entangled with their neighbors to create a contiguous mechanically interlocked structure. Graphical modeling further describes the intricate geometry and unique features of the structure. A modification to classical laminate analysis is necessary to capture the complex progressive twist and tilt rotations of the laminae. Parametric calculations of the stiffness and strength of the structure are presented, including analysis of its degree of isotropy and discussion of its functionality.

## Results

**BLU structures in the tarsus cuticle**. Transversal and longitudinal cross-sections through the tarsus of the *Scorpio Maurus Palmatus* were imaged by SEM (Fig. 1a–d). The region of interest was the endocuticle (Fig. 1e, f), the innermost cuticle layer which consists of three-dimensional BLUs architecture. Sample preparation and imaging techniques are detailed in Methods.

Magnified views of the transversal ($yz$) and longitudinal ($xz$) cross-sections are shown in Fig. 2. The endocuticle consists of about 20 layers, the thickness of which gradually decreases from about 8 μm in the outermost layer to about 3 μm in the innermost layer[26]. Each layer houses a two-dimensional periodic array of tightly packed BLUs. In the following we will use the prefix *intra* to denote features between BLUs in a given layer, and *inter* to denote features between BLU layers. The BLU layers are bound to each other by thin interfacial layers ('interlayers') of chitin fiber bundles (Fig. 2a, b), each about 1.5–2 μm thick. The fibers

orientation in the interlayers is predominantly in the direction of the tarsus circumference (*y*-axis), seen in the bright unidirectional interlayers in Fig. 2c. The BLUs spatial period in the middle layers, that is the distance between the centers of neighboring BLUs, is about 1.5–2 μm in both the transversal and longitudinal directions (see dimensions on Fig. 2c, d). Each BLU is separated from its neighbors in both the *x* and *y* directions by thin fibrous interfacial layers ('intralayers') about 100–200 nm thick (Fig. 2e, f), whose fibers are parallel to the BLU layers plane (*xy* plane).

The BLU itself consists of about 40–100 laminae, each about 50–100 nm thick and comprising a single layer of unidirectional chitin fibers embedded in a proteinaceous matrix. These estimates were obtained by counting laminae in SEM images (Fig. 2) and dividing the BLU height (in *z*-direction) accordingly, as well as by directly measuring the fibers diameter in TEM images (Fig. 3d), both indicating that a lamina thickness is comparable to the fiber diameter. The BLU shape is roughly that of a left-handed helical prism, generated by progressively rotating ('twisting') laminae and completing half a revolution per BLU. Thus, the total twist angle is about 180°, and the angular increment of each lamina is about 1.8–4.5° depending on the number of laminae.

From our observations we conclude that the twisting rotation axis is parallel to the *z*-axis and is located at the laminae corners ('off-axis'), rather than at the laminae centers as commonly reported (e.g.,[12–14]). The fibers orientation in a lamina can be inferred from its texture: using Fig. 2c for instance, when the texture is smooth and grooved by long lines, as seen in the upper-right region of the BLUs, the fibers are observed from their sides; when the texture is rough and uneven, as seen in the mid-region of the BLUs, the fibers are observed from their edges. This is also visible in the BLU in the middle of Fig. 2f, where we see fibers from both their sides and edges. Thus, in the midplane lamina, located at the BLU mid-height, the fibers appear to be oriented longitudinally in the *x* direction; for example, in Fig. 2c they are roughly perpendicular to the image plane. Thus, at the interlayers, after completing a ±90° turn, the lamina fibers are oriented in the *y* direction, coaligned with the interlayers' fibers.

The BLU's helical turn restarts its twist at each layer of BLUs: a −90° twist angle at a layer top resets to +90° when moving in the *z* direction across an interlayer to the layer above it; this means that the helicoid twist is discontinuous, with a 180° jump, when moving from a layer to its adjacent next. Thus, the helicoid is not infinite in the mathematical sense.

The upper and lower BLU laminae merge into the interlayers, forming a continuous fibrous transition between layers. This is clearly seen in both views, particularly in the oblique view of Fig. 2f which shows the merging of laminae of several nested BLUs into an interlayer at their top; the interlayer appears as 'flowing' down toward the right-bottom corner of the image, and its fibers are unidirectional. TEM imagery (see Methods) provides additional insight on the merging feature, showing continuous transition between an interlayer and the layers of BLUs above and below it (Fig. 3b, c), and the 'flow' of fibers from BLUs on both sides into an interlayer (Fig. 3d). Similarly, fibers from neighboring BLUs merge laterally into the intralayers separating them (see for example Fig. 3a). Note the white oblong dots in Fig. 3b, c, which are slant cross-sections through fibers running inside the cuticle pore canals. The pore canals, also seen in the SEM images in Fig. 2c (side view) and Fig. 2f (cross-section), are vertical tubular ducts that function as a material transport system[24,26].

Another so-far unnoticed structural feature also appears from our observations: in addition to the twist angle, the laminae are progressively rotated upward and downward ('tilted') out-of-plane, reminiscent of a handheld fan. This is seen for example in Fig. 2c, e, where the midplane lamina of a BLU is roughly parallel

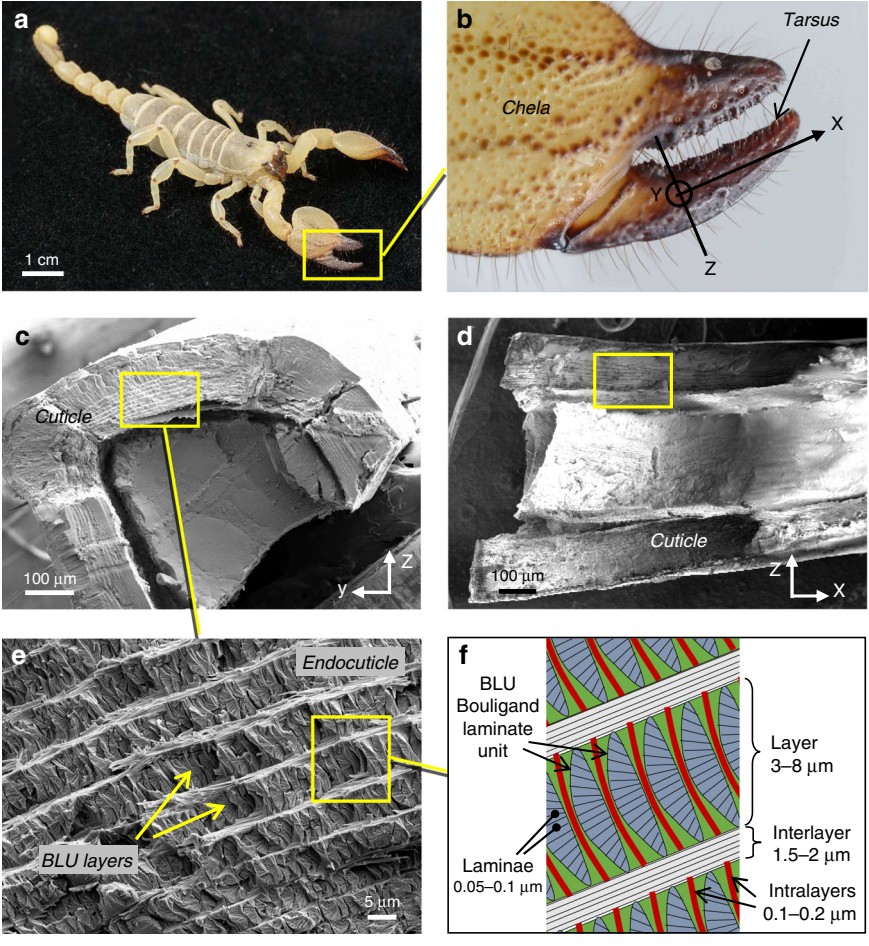

**Fig. 1 Cross-sections through the scorpion tarsus. a** *Scorpio Maurus Palmatus*. **b** Chela (pincer), tarsus (moveable claw) and axes definition. **c** YZ transversal cross-section, focusing on the lower part of the tarsus. **d** XZ longitudinal cross-section, focusing on the upper part of the tarsus. **e** SEM magnification of the *yz* cross-section showing the endocuticle and BLU (Bouligand laminate unit) layers. **f** Definition of terminology.

to the *xy* plane, whereas the laminae close to the top and bottom of the BLU tend to be perpendicular to the *xy* plane. Thus, the tilting angle starts from 0° at the midplane lamina and reaches about ±90° turn close to the interlayers. Accordingly, the total tilt angle is about 180°, and the tilting angular increment of each lamina is about 1.8°–4.5° depending on the number of laminae, approximately the same as that of the twisting angle. The tilting axis is orthogonal to the *z*-axis (that is, parallel to the *xy* plane), and is located within the plane of each lamina, coinciding with the axis of the innermost fiber (the fiber that intersects the twisting axis). Thus, the tilting axis is a local axis within a lamina, running through one of its edges. Because all the fibers in the lamina seem to lie parallel to each other (see for example the exposed laminae edges in Fig. 2f), they remain parallel to the *xy* plane regardless of the twist and tilt angles. Note that the orientation of the *xy* plane and the *z*-axis is globally fixed with respect to the samples, as shown in Figs. 1 and 2, and is not affected by either twisting or tilting.

The arrangement of BLUs in a layer is that of a compact 2D array, with a similar spatial period in both the *x* and *y* directions (Fig. 2c, d), resembling a lattice with square cells each occupied by a single BLU. All BLUs in a layer, as well as across layers, are facing the same direction; in other words, their orientation around the *z*-axis is the same, so that the mid-lamina fibers are always aligned with the *x* direction. The different patterns seen in the transversal (Fig. 2a, c) and longitudinal (Fig. 2b, d) views reflect the asymmetric nature of the BLU around the *z*-axis.

Because the twisting axis is at the laminae corners and not at their centers, combined with the fact that it completes only half a turn, the BLU shape does not possess symmetry around the twisting axis (nicely seen in Fig. 2f). Therefore, in the transversal views we see the repeating distinct shape resembling a parabolic segment, whereas in the longitudinal views we see a repeating warped shape. These traits will become clear in the next section where the BLU geometric model is described.

Last but not least, the warped shape and the common orientation allow the BLUs to nest in tight fit, in other words to embrace their neighbors and be embraced by them, thereby creating a self-assembled interlocked puzzle.

**Geometric model.** In summary, the observed structural features are (Fig. 4): (a) laminae are reinforced by unidirectional fibers; (b) laminae are progressively twisted within a ±90° range around an axis at their corners (*z*-axis); the twist angle resets to −90° with each new BLU layer; (c) laminae are progressively tilted within a ±90° range around the innermost fiber axis (1-axis); (d) fibers in the midplane lamina are oriented longitudinally (*x*-axis) at 0° twist; (e) BLUs in a layer are nested in a periodic, tightly packed, interlocked 2D array; (f) fibers in end laminae are oriented transversely (*y*-axis) and merge into interlayers which separate BLUs layers; and (g) neighboring BLUs are separated by an intralayer.

These observations are captured and clarified in the following geometric model. The BLU is a laminate, built up from *n* laminae

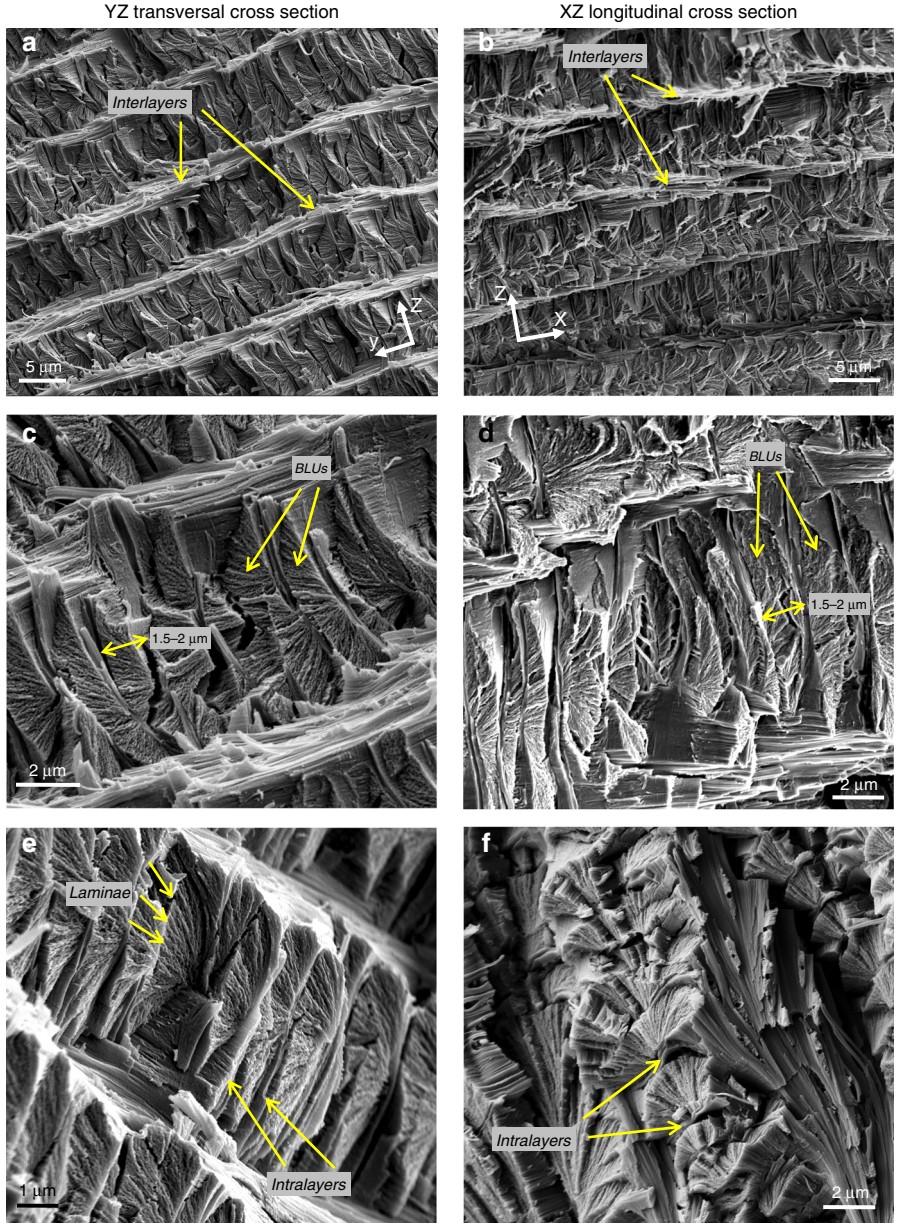

**Fig. 2 SEM transversal and longitudinal cross-sections through BLU (Bouligand laminate unit) layers. a, b** BLU layers and interlayers. **c, d** Individual BLUs and spatial period in $x$ and $y$. **e** Intralayers between BLUs. **f** Oblique view of nested BLUs.

laid one on top of the other. The lamina local coordinate system (1,2), the laminate global coordinate system $(x,y)$ and geometrical parameters are defined in Fig. 5a. The midplane lamina is positioned so that it is in the $xy$ plane and its fibers are oriented in the $x$ direction. Each lamina is rotated by a twist angle $\theta$ and a tilt angle $\phi$ with respect to the midplane. The twisting axis is $z$, located at the lamina corner. The tilting axis is the lamina local 1-axis, which coincides with the centerline of the lamina innermost fiber. Both the global $(x,y)$. and local (1,2) coordinate systems are right-handed, and therefore both rotation angles are defined by the right-hand rule. As observed in the scorpion endocuticle, the helicoid is left-handed and therefore $\theta$ decreases with rising $z$, ranging from 90° at the bottom lamina to −90° at the top lamina, whereas $\phi$ increases with rising $z$, ranging from –90° to 90°, respectively. Numbering the laminae from 1 at the bottom to $n$ at the top, the twist and tilt angles of lamina $k$ are $\theta_k$ and $\phi_k$, respectively. Because the angular span of both twist and tilt is the

same, and as the angular increments are nearly uniform, the observations lead to $\phi_k = -\theta_k$.

As observed, each lamina consists of a single layer of tightly packed unidirectional chitin fibers embedded in a proteinaceous matrix, and we may therefore assume that the lamina thickness $t$ is equivalent to the fiber diameter. The matrix fills the gaps between fibers in a lamina and the tilting gaps between laminae. The $z$-position of the lamina inner edge, $z_k$, is determined by the cumulative thickness of the laminae below it, and the lamina twist and tilt angles are determined by the cumulative angular increments between the laminae below it (Supplementary Note 1):

$$z_k = \frac{t}{2}(2k - n - 1)$$
$$\theta_k = -\phi_k = \frac{\pi}{2} - \frac{k-1}{n-1}\pi \tag{1}$$

where $t$ is assumed to be uniform.

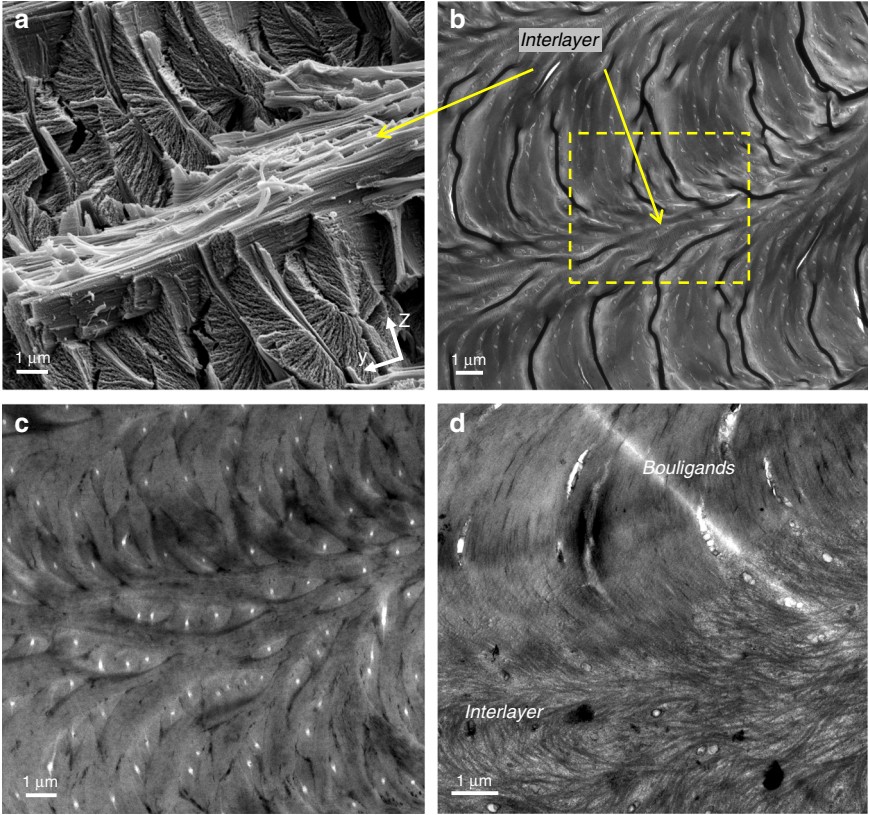

**Fig. 3 Merging of BLU (Bouligand laminate unit) fibers into the interlayer.** YZ transversal view. **a** SEM image of an interlayer separating two BLU layers, and **b**, **c** TEM images of the same region. The tiny white dots are cross-sections through the pore canal fibers. **d** Magnified TEM image of the squared region showing fibers merging into an interlayer.

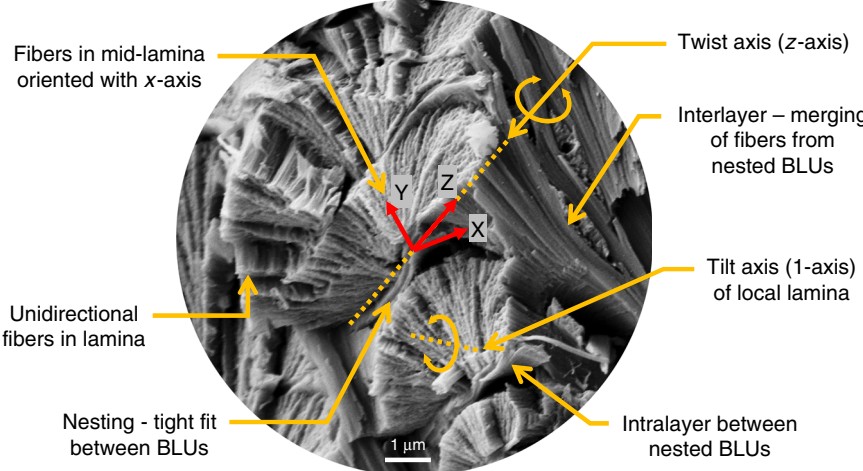

**Fig. 4 Summary of observed structural features.** Oblique view of an XZ longitudinal cross-section containing BLUs (Bouligand laminate units).

A lamina $k$ is depicted by translating the midplane lamina to position $z_k$, and applying an angular transformation with the angles $\theta_k$ and $\phi_k$. To create the graphical model presented in Fig. 5, this transformation was repeated for all $n$ laminae (see Supplementary Note 1 and Supplementary Movie). The images generated by the model convincingly compare with their biological counterparts, as shown in Fig. 6.

Because BLUs are tightly packed, the lamina (fiber) length $l$ and width $w$ (Fig. 5a) can be considered as equivalent to the BLUs spatial period in the respective directions. As the period was observed to be roughly the same in both in-plane directions (Fig. 2c, d), we may approximate the lamina as a square ($w = l$).

Also, because the lamina thickness is equivalent to the fiber diameter, the number of fibers in a lamina is $w/t$.

**Nesting and interlocking**. As a consequence of rotation around a vertical axis at the lamina corner instead of its center, the resulting model has no apparent symmetry about any of the three major planes, nor around any of the major axes. This is seen in the four main views in Fig. 5d–g. Furthermore, the effect of tilting is different in the upper half of the BLU compared to its lower half, as manifested by the narrower cross-section in the latter. This narrowing occurs as a result of geometrical interference

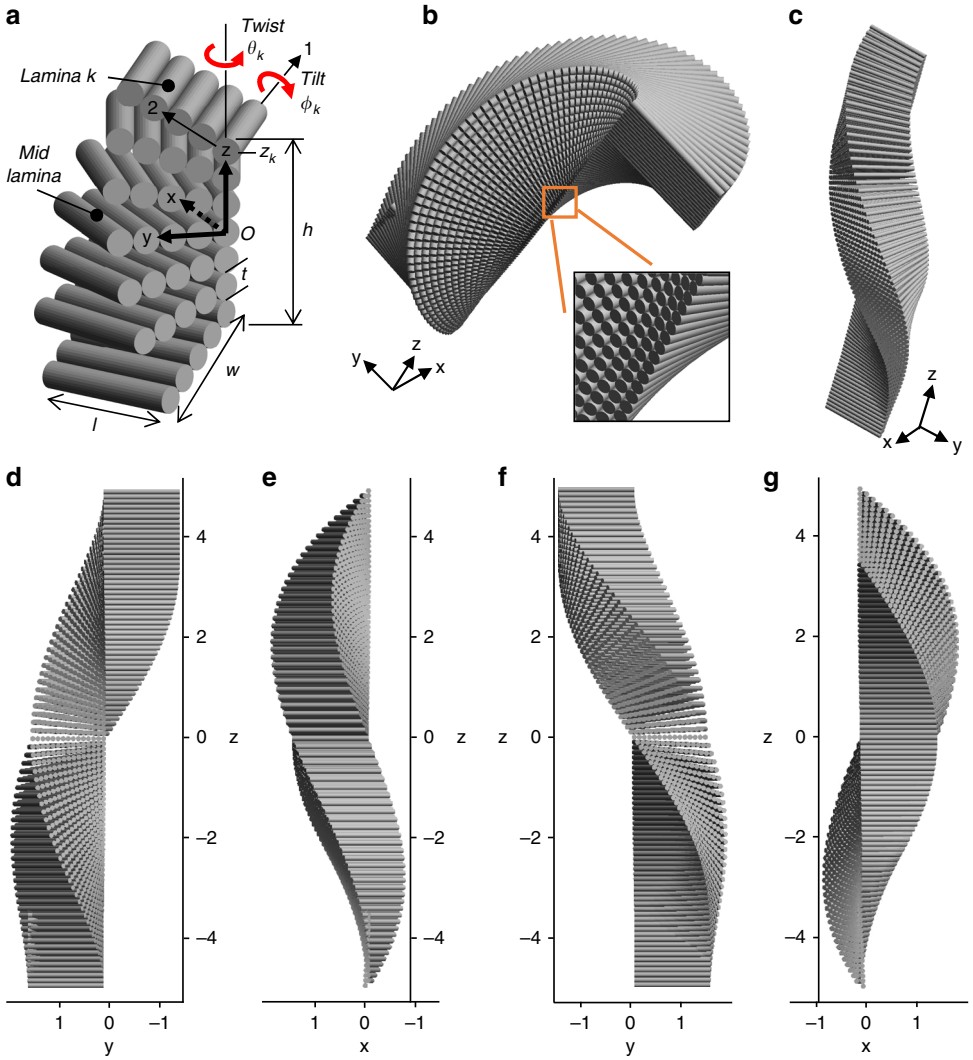

**Fig. 5 BLU (Bouligand laminate unit) geometrical model. a** Coordinate systems – layer/laminate ($x,y$) and lamina (1,2); lamina $k$ twist angle is $\theta_k$, tilt angle is $\phi_k$, and $z$-position is $z_k$; $l$ and $w$ are lamina length and width. **b**–**g** Views of modeled BLU. The parameters are: number of laminae $n = 71$, lamina thickness $t = 100$ nm, lamina size $l = w = 1.5\,\mu$m, range of twist angle $\theta = \pm 90°$ left turn, range of tilt angle $\phi = \mp 90°$, angular increment $\delta_\theta = -\delta_\phi = -2.54°$, height (excluding tilt contribution) $h = 7.1\,\mu$m.

between laminae; evidently, this is impossible in the biological tissue in which these laminae are rearranged in such a way as to avoid interference. Indeed, careful observation of the images in Fig. 2 reveals a variance between the upper and lower regions of a layer, which might be caused by this inherent geometrical feature.

*xy* cross-sections through the BLU model (Fig. 7d, e), equally distanced from the midplane, reveal horizontal layers with fan-shaped spreading of fibers, each belonging to a lamina with a different twist angle. The fibers are short and discontinuous between BLUs, and are all parallel to the *xy* plane. The angular span of the fibers in such a layer can be calculated (see Supplementary Note 2), and is the same for the upper and lower cross-sections ($\theta_k - \theta_i = 29.3°$ in the example in Fig. 7d, e), even though their shapes are different. Mathematically, the helicoid does not have helical symmetry, as its appearance is not constant along the *z*-axis, demonstrated by the very different shape of these two cross-sections. This asymmetry can also be seen by tracing fiber end points in the model (Fig. 5), which form a loosely helical shape that does not make a constant angle with the twisting axis.

The geometric model provides insight on the arrangement and matching of many BLUs (Fig. 7a–c). The off-axis helicoidal shape

of each BLU matches the counter-shape of its neighbors, resulting in a tight fit between BLUs. The BLUs mesh into each other much like a jigsaw puzzle or a gear train, generating shear interlocking in all three shear planes (see illustration in Fig. 7c). This interlocking prevents in-plane shear (*xy* plane), as well as out-of-plane shear (*xz* and *yz* planes) and BLU spin (rotation around *z*). Interlocking is achieved by the wavy shape of the interfacial surfaces between BLUs (Fig. 7f), such that when an external shear stress is applied, the displacement of BLUs is mechanically precluded. This mechanism is also effective when an external tensile stress is applied, such that the pullout of BLUs is prevented in all directions by the topological obstacles set by their neighbors. Further stiffening of the BLUs array is secured by the inter- and intra-layers whose fibers merge into the BLUs. Shear interlocking was observed in other natural microstructures such as the dovetailed platelets of nacre[27], preventing shear in just a single direction, but the shear interlocking presented here is three-dimensional. This 3D interlocking could serve toughness by diverting cracks at the varying angular directions of the laminae and fibers and at the inter- and intra-layers. More importantly, it could serve stiffness and strength of the tarsus, as the thin

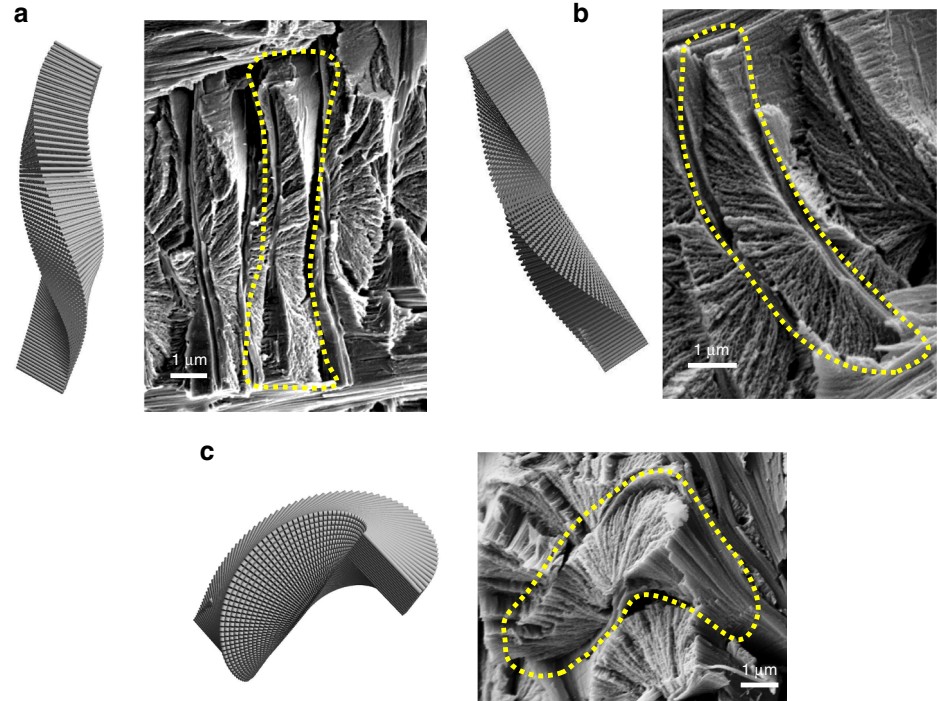

**Fig. 6 Comparison between the model and its biological counterpart. a** Longitudinal cross-section. **b** Transversal cross-section. **c** Longitudinal cross-section, oblique view.

macrostructure of the tarsus is sensitive to in-plane shear stresses caused by external torsion around $x$ and to out-of-plane shear stresses caused by external forces in the $y$ or $z$ direction.

**Laminate elastic modeling**. The BLU laminate conformation has three unique geometrical features: (i) laminae are rotated off-axis, that is around an axis at their corners rather than centers, generating a warped structure in which distant laminae are not overlapping (when viewed in the $z$ direction); (ii) laminae are rotated around two axes simultaneously, twist and tilt, such that successive laminae are not parallel and the angle between the planes of distant laminae is large; (iii) the laminate lateral dimension is smaller than its height, and the BLUs are nested in each other. Theoretical analysis of this unique laminate type is important for understanding the rationale behind this structure and the benefits that might be associated with it. We propose a model that uses classical laminate theory[28], with proper adaptations for incorporating the effect of lamina tilting. Our approach is explained and justified in Supplementary Note 3. Presented here is just the result of the laminate stiffness calculation, whereas the modeling details are found in Supplementary Notes 4 and 5.

Classical laminate theory assumes that the laminate $x$ and $y$ dimensions (lateral width) are much larger than its $z$ dimension (thickness or height)[28], a condition that is not met within the scale of a single BLU. However, shear interlocking between BLUs, as well as the thin fibrous intralayers connecting BLUs, ensure that in-plane stresses are effectively transferred between BLUs, making the BLUs layer structurally contiguous. This can be observed in the in-plane cross-sections in Fig. 7d, e, in which the short fibers from multiple BLUs form a wide, continuous, nearly unidirectional lamina.

The BLU laminate is of the balanced antisymmetric type, meaning that it consists of pairs of identical $\theta_k$ and $-\theta_k$ laminae, arranged at equal $z$-distance from the midplane. When the laminate is under load, the forces **N** and moments **M** acting on it

are related to the laminate plane strains $\boldsymbol{\varepsilon}^0$ and curvatures $\boldsymbol{\kappa}$ by

$$\begin{bmatrix} \mathbf{N} \\ \mathbf{M} \end{bmatrix} = \begin{bmatrix} \mathbf{A} & \mathbf{B} \\ \mathbf{B} & \mathbf{D} \end{bmatrix} \begin{bmatrix} \boldsymbol{\varepsilon}^0 \\ \boldsymbol{\kappa} \end{bmatrix} \tag{2}$$

where $\mathbf{N}$, $\mathbf{M}$, $\boldsymbol{\varepsilon}^0$ and $\boldsymbol{\kappa}$ are vectors comprising components in the $x$, $y$ and $s$ (that is, shear or $xy$) directions. $\mathbf{N}$ and $\mathbf{M}$ are given per unit length of the laminate, in the $x$ or $y$ direction. The 6x6 laminate stiffness matrix consists of the $3 \times 3$ stiffness matrices $\mathbf{A}$, $\mathbf{B}$ and $\mathbf{D}$ given by

$$\begin{aligned}
\mathbf{A} &= t \sum_{k=1}^{n} \mathbf{Q}_k \\
\mathbf{B} &= t \sum_{k=1}^{n} \mathbf{Q}_k \left( z_k + \frac{1}{2} w \sin \phi_k \right) \\
\mathbf{D} &= t \sum_{k=1}^{n} \mathbf{Q}_k \left( z_k^2 + z_k w \sin \phi_k + \frac{1}{3} w^2 \sin^2 \phi_k \right)
\end{aligned} \tag{3}$$

where $t$ is the lamina thickness, assumed to be uniform and thin, $w$ is the lamina width (Fig. 5a), $n$ is the number of laminae in the laminate, $k$ is a specific lamina ($k=1..n$), $z_k$ is the $z$-position of the inner edge of lamina $k$, and $\phi_k$ is its tilt angle (both defined in Eq. (1) and Fig. 5a). $\mathbf{Q}_k$ is lamina $k$ stiffness matrix, obtained by transforming the lamina stiffness matrix from its material axes (1,2) to the BLU principal axes $(x,y)$, using the twist angle $\theta_k$. A lamina is treated as a composite, consisting of tightly packed unidirectional chitin-protein fibers embedded in a proteinaceous matrix. The lamina stiffness calculation uses rules of mixtures and includes a correction for the limited fiber length $l$ (see details in Supplementary Note 4).

Because of the tilt rotation, the strain and stress at a point in a tilted lamina depend on the point's specific $z$-position. Consequently, the assumption of laminate theory that the stress is uniform throughout a lamina cannot be applied, and the stress variation within a lamina has to be accounted for. This

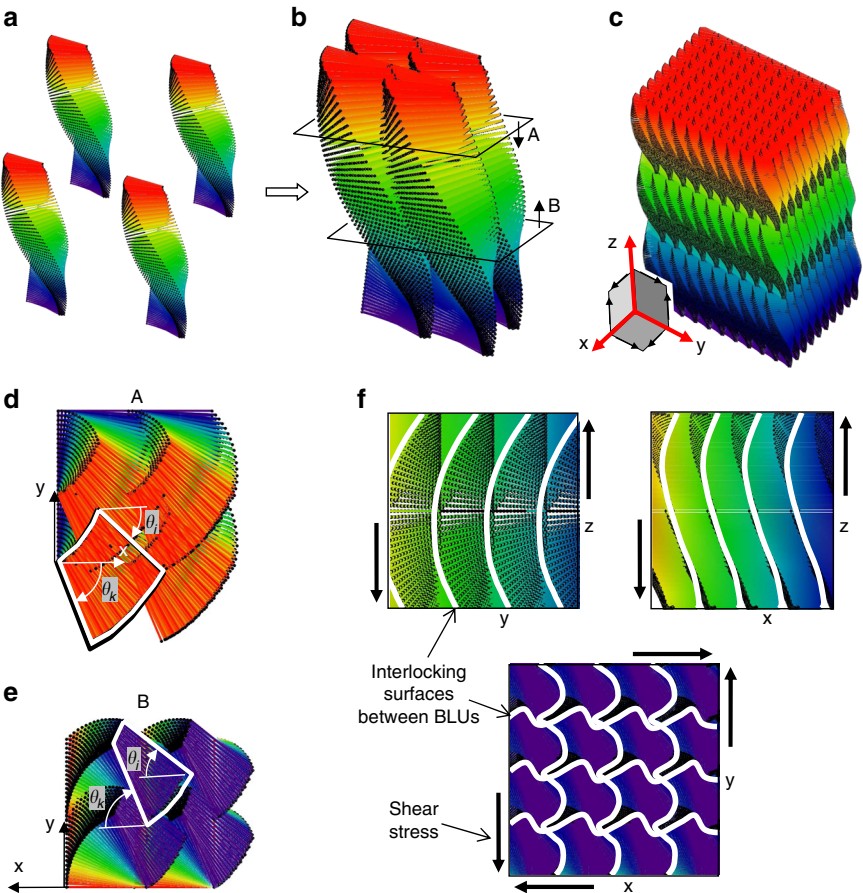

**Fig. 7 BLUs (Bouligand laminate unit) arrangement. a, b** Compact packing of 4 BLUs. Cross-sections A and B at $z = \pm 2\,\mu m$ are indicated. **c** View of layers arrangement and shear interlocking planes. **d, e** Cross-sectional views A and B; horizontal layers are indicated. **f** Shear interlocking surfaces prevent displacement of BLUs under shear and tensile stresses. The parameters are the same as in Fig. 5, except for: lamina thickness $t = 75$ nm, lamina size $l = w = 1.35\,\mu m$, height (excluding tilt contribution) $h = 5.3\,\mu m$.

modification is expressed in Equation (3) by the terms containing $\phi_k$, whose impact is higher toward the upper and lower regions of the BLU (larger $|\phi_k|$). The solution reduces to the classical form for thin uniform laminae when there is no tilt ($\phi_k = 0$). Note that, unlike a laminate with parallel laminae, the forces and moments per unit length are functions of the lamina width $w$. The incorporation of tilting in the model causes the stiffness matrix to be very different from classical laminates with parallel laminae, deeply affecting the laminate elastic properties as calculated ahead. Tilting leaves small angular gaps between adjacent laminae as a result of the difference in their tilt angle. These gaps, seen in Fig. 2, are assumed to be filled by matrix, and their contribution to the BLU stiffness is negligible and therefore ignored in the analysis; however, the presence of matrix in these gaps is essential to ensure stress transfer between laminae.

**Laminate elastic properties and isotropy.** The BLU effective elastic properties are calculated from the laminate stiffness matrix (see description in Supplementary Note 6). The calculation uses the average laminate stresses, $\bar{\boldsymbol{\sigma}} = \mathbf{N}/H$, where the overall height of the BLU due to tilting, $H = h + 2(w - t)$, is used rather than the net height without the contribution of tilting, $h = nt$. The laminate has a total of 9 engineering constants: three elastic moduli, two Poisson's ratios, and four shear coupling coefficients. In addition, the three effective flexural stiffnesses (moment / curvature), two bending and one torsional, are calculated.

Estimated material properties and calculated fiber, lamina and laminate elastic constants are summarized in Table 1. It should be kept in mind that these are just ballpark figures, as the components data is scarce and dependent on conditions. However, as our goal is the understanding of the underlying features and benefits of the BLU conformation, the exact material properties are not critical. The BLU composite is a hierarchical structure consisting of four assembly levels[26]: (1) α-chitin chains packed in ~5 nm chitin-protein fibrils; (2) fibrils collected into 50–100 nm chitin-protein fibers; (3) fibers embedded in protein and compactly packed in laminae; and (4) plied laminae assembled in the BLU. In Table 1, we use the subscripts c and p to denote the α-chitin chains and the protein, respectively, the subscript f to denote the chitin-protein fibers, the subscripts 1,2 to denote the lamina, and the subscripts $x,y,s$ to denote the laminate (BLU). The three higher hierarchical levels (2–4) – fiber, lamina and laminate – are represented in Table 1, including their respective volume fractions: $V_c$ chitin in a fiber, $V_f$ fibers in a lamina, and $V_1$ laminae in the laminate. Assuming square arrangement in a lamina, $V_f = \pi/4$, the ratio between the fiber cross-sectional area and the square inscribing it. The calculated value of $V_1$ accounts for the gaps between laminae due to tilt, and is the ratio between the cross-sectional areas lamina/(lamina +gap), or $(1 - w\delta_\phi/2t)^{-1}$.

Typical material properties are provided in the recent review by Politi et al.[24], including: (1) the elastic modulus of the crystalline α-chitin is in the range of $E_c = 88$–120 GPa, based on simulations,

**Table 1 Estimated material properties and calculated elastic constants for fiber, lamina and laminate[a].**

| Estimated material properties | | | | Calculated fiber constants[b] | | | | Calculated laminate constants[c,d] | | |
|---|---|---|---|---|---|---|---|---|---|---|
| $n$ | number of laminae | 100 | | $E_{1f}$ | tensile modulus | 70.4 | | $\bar{E}_x$ | long. modulus | 9.6 (14.7) |
| $t$ | fiber, lamina thickness | 75nm | | $E_{2f}$ | trans. modulus | 4.8 | | $\bar{E}_y$ | trans. modulus | 6.6 (9.8) |
| $l,w$ | lamina length, width | 2 µm | | $G_{12f}$ | shear modulus | 1.7 | | $\bar{G}_{xy}$ | shear modulus | 2.0 (3.1) |
| $V_c$ | chitin/fiber fraction[e] | 0.70 | | $v_{12f}$ | Poisson's ratio | 0.4 | | $\bar{v}_{xy}$ | Poisson's ratio | 0.10 |
| $V_f$ | fibers/lamina fraction[e] | 0.78 | | $v_{21f}$ | Poisson's ratio | 0.03 | | $\bar{v}_{yx}$ | Poisson's ratio | 0.07 |
| $V_1$ | laminae/laminate frac[e] | 0.70 | | **Calculated lamina constants[b]** | | | | $\bar{\eta}_{ij}$ | shear coupling | 0 |
| $E_c$ | chitin modulus | 100 | | $E_1$ | long. modulus | 41.8 | | $\bar{K}_x$ | bend stiffness | 0.55 (0.30) |
| $v_c$ | chitin Poisson's ratio | 0.4 | | $E_2$ | trans. modulus | 3.3 | | $\bar{K}_y$ | bend stiffness | 2.9 (1.7) |
| $E_p$ | protein modulus | 1.5 | | $G_{12}$ | shear modulus | 1.2 | | $\bar{K}_s$ | torsion stiffness | 0.35 (0.19) |
| $v_p$ | protein Poisson's ratio | 0.4 | | $v_{12}$ | Poisson's ratio | 0.40 | | | | |
| | | | | $v_{21}$ | Poisson's ratio | 0.03 | | | | |

[a]Units: elastic moduli are in GPa; flexural stiffnesses are in N µm²
[b]Fiber and lamina elastic constants are in principal material axes (1,2)
[c]Laminate elastic constants are in principal laminate axes (x,y), and are effective (average) laminate values
[d]Values in parenthesis are without tilting, for comparison
[e]The product of these volume fractions yields the chitin fraction in the entire structure, $V_c V_f V_l \cong 0.38$

and the value of 100 GPa is typically used; (2) the elastic modulus of the proteinaceous matrix is in the range of $E_p = 0.01-1$ GPa, estimated by indirect experimental and theoretical methods; (3) the typical Poisson's ratio for the chitin and protein is in the range of $v_c, v_p = 0.3-0.45$; and (4) the chitin volume fraction in the entire composite is in the range of $V_c = 0.1-0.4$. These values were used as guidelines in our analysis, except for $E_p$ for which 1.5 GPa was used, assuming the protein is partially reinforced by ion metals and chitin fragments. We used this value as an effective matrix stiffness that assures stress transmission to the short fibers of the BLU, which is degraded by a significant length correction factor, $\eta_1 = 0.75$, according to shear-lag theory (see details in Supplementary Note 4).

It is instructive to compare the results of the calculated tensile moduli to nanoindentation measurements of the tarsus endocuticle of the *Scorpio Maurus Palmatus*[26], which yielded $E_x = 8.5$ GPa and $E_y = 8.2$GPa (dry samples). These measurements are an endocuticle average over the BLUs, intralayers and interlayers covered by the indenter, and their values tend to equalize due to the inherent irregularities of the measured surfaces. In particular, to isolate the value of the BLU alone from the endocuticle average, $E_x$ should be increased because the stiffness of the interlayer in this direction, $\sim E_2$, is much lower than that of the BLU; similarly, $E_y$ should be decreased because the stiffness of the interlayer in this direction, $\sim E_1$, is much higher than that of the BLU. Thus, the calculated values, $\bar{E}_x = 9.6$ GPa and $\bar{E}_y = 6.6$ GPa (Table 1), are generally in agreement with the measurements. Moreover, the calculated values reveal the BLU anisotropy not captured by the measurements because of the averaging effect described above. Conversely, we can say that the elastic moduli achieve isotropy only at the higher assembly level of the endocuticle, which encompasses the BLUs, intralayers and interlayers, such that the lower $\bar{E}_y$ at the single BLU level is compensated by the higher interlayer stiffness $E_1$ at the layer +interlayer level.

We see that the tensile moduli in both the $x$ and $y$ directions are fairly different, and in that respect the BLU is only barely quasi-isotropic, but at the same time they indicate a much higher degree of isotropy than that of a single lamina. The shear-normal cross coupling constants $\bar{\eta}_{ij}$ are null, indicating shape stability under load. The bending stiffness $\bar{K}_y$ (around $x$) is much higher than $\bar{K}_x$ (around $y$), as the fibers far from the midplane are more aligned with the $y$ direction and their contribution to the bending stiffness is predominant. These trends are demonstrated in

Fig. 8a–c, which show the elastic constants as functions of the transformation angle $\psi$ (around $z$-axis) of the coordinate system $x'y'$ with respect to $xy$. It is seen that at intermediate values of $\psi$, the anisotropy may even grow as a result of the larger asymmetry of the BLU with respect to the $x'y'$ system; for example, $\bar{E}_{x\prime}$ has a minimum at $\psi \cong 50°$ (Fig. 8a), and $\bar{\eta}_{ij}$ deviate significantly from 0 (Fig. 8c). Note that in these plots, at $\psi = 0$ the subscripts $x'$ and $y'$ can be replaced by $x$ and $y$, respectively ($E_{x\prime} = E_x$, etc.), whereas at $\psi = 90°$ they can be replaced by $y$ and $x$, respectively ($E_{x\prime} = E_y$, etc.).

The degree of isotropy deeply depends on the range of the total twist angle $2\Theta$ (Fig. 8d), where, as expected, in a hypothetical full 360° turn the elastic moduli and bending stiffness achieve almost perfect quasi-isotropy; at the observed 180° half-turn the elastic moduli barely approach quasi-isotropy, whereas the bending stiffness is very far from it. Note that in the twist range ~110°–180° the isotropy trend of the tensile moduli is rising whereas that of the shear modulus and bending stiffness is falling, so that a 180° twist favors the tensile moduli over the other constants. Finally, it is seen that the degree of isotropy is not affected by tilting (Fig. 8a, b). Evidently, the in-plane isotropy (or close to it) is an important property of the structure, as the direction of the external loads on the tarsus cannot be predicted, whereas the bending stiffness clearly is higher for bending around the $x$ axis.

The effect of tilting can be observed by comparing the laminate to an identical hypothetical un-tilted laminate. Tilting decreases the in-plane elastic constants significantly (for example, $\bar{E}_x$ decreases from 14.7 GPa to 9.6 GPa), while increasing the flexural elastic constants significantly (for example, $\bar{K}_y$ increases from 1.7 N µm² to 2.9 N µm²) (Table 1). This is also shown in Fig. 8a, b as a function of the transformation angle. The reason for these trends is that tilting shifts fibers farther away from the midplane, redistributing the reinforcement material without changing its overall amount. Thus, the overall height of the structure is increased at the expense of reducing the reinforcement density, seen in the laminae/laminate volume fraction, $V_1 = 0.7$ (Table 1), which without tilting would have been 1. Consequently, the structure is stiffer under flexural loads but less stiff under in-plane stresses, a tradeoff made possible by tilting. Obviously, high flexural stiffness is important in order to resist local contact pressure exerted on the cuticle.

In terms of the scorpion biomechanical loading environment, its chela needs to withstand high clamping loads when attacking

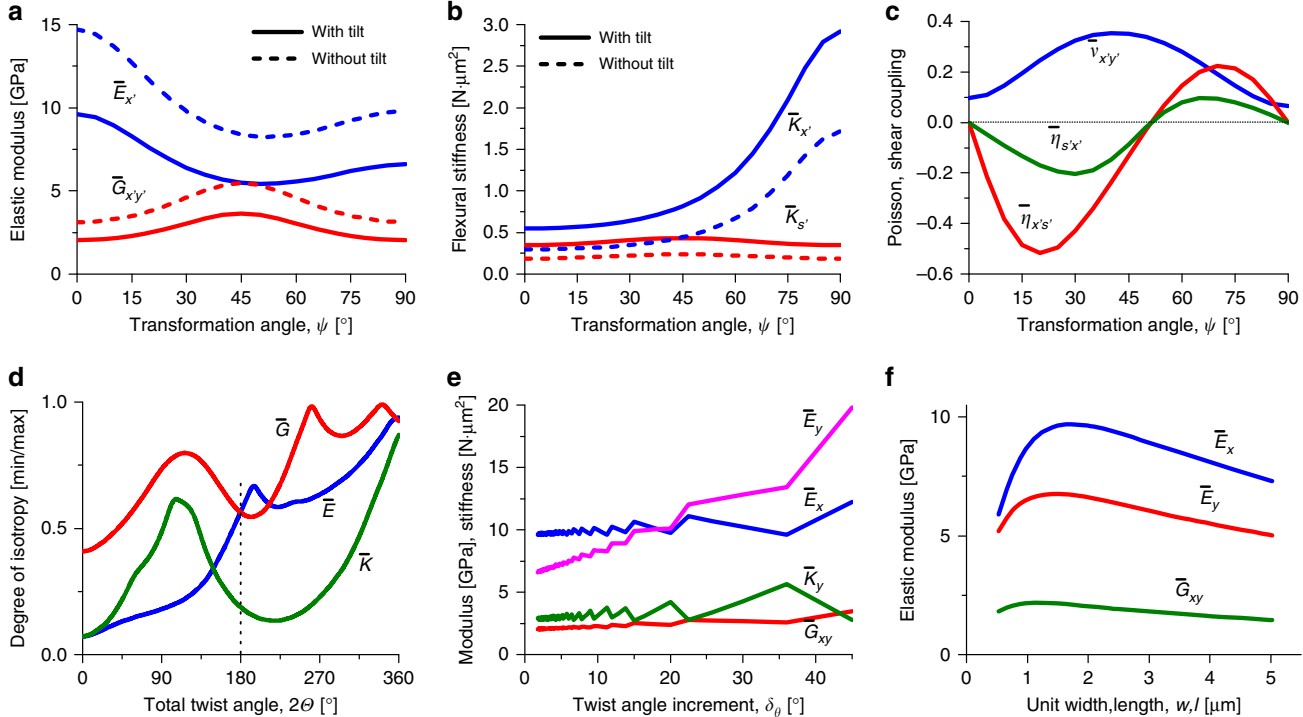

**Fig. 8 BLU (Bouligand laminate unit) laminate analysis. a–c** Effective elastic moduli (tensile $\bar{E}_{x'}$ and shear $\bar{G}_{x'y'}$), flexural stiffnesses (bending $\bar{K}_{x'}$ and torsion $\bar{K}_{s'}$), Poisson's ratio ($\bar{\nu}_{x'y'}$) and shear coupling coefficients ($\bar{\eta}_{x's'}$ and $\bar{\eta}_{s'x'}$), vs. the transformation angle $\psi$ (around $z$-axis) of the coordinate system $x'y'$ with respect to the principal system $xy$. **d** Degree of isotropy (the ratios $\bar{E}_{min}/\bar{E}_{max}$, $\bar{G}_{min}/\bar{G}_{max}$ and $\bar{K}_{min}/\bar{K}_{max}$) vs. the total twist angle of the laminate $2\Theta$. **e** Effective elastic moduli and bending stiffness vs. the twist angle increment $\delta_\theta$ at constant BLU height. **f** Tensile and shear elastic moduli vs. the laminate width $w$ or length $l$ ($w = l$). The nominal parameters are as in Table 1: number of laminae $n = 100$, lamina thickness $t = 75$ nm, lamina size $l = w = 2$ µm, range of twist angle $\theta = \pm 90°$ left turn, range of tilt angle $\varphi = \mp 90°$, angular increment $\delta_\theta = -\delta_\varphi = -1.80°$, height (excluding tilt contribution) $h = 7.5$ µm.

prey, and at the same time high contact loads when defending against threats. The first type of load is met by the thick tubular structure of the tarsus, which provides a high bending stiffness at the structural level above the cuticle itself. Thus, a high tarsus stiffness can be achieved even with a relatively low cuticle tensile stiffness. The second type of load is met by the bending stiffness of the cuticle itself against contact pressure. Thus, a higher cuticle bending stiffness at the expense of a lower tensile stiffness, as demonstrated by the model, substantiates a reasonable biomechanical tradeoff.

The elastic constants are also dependent on the twist angle increment $\delta_\theta$, as demonstrated in Fig. 8e for constant BLU height. An interesting finding is the BLU width (or length) dependence of the elastic moduli (Fig. 8f), which exhibits optimal (maximum) values in the region of $w = l \cong 1 - 2$ µm, in correspondence with the experimental observations. This is the result of opposing parametric trends: (1) a modulus increase when the fibers are longer (larger $l$), due to better matrix-fiber stress transfer; and (2) a modulus decrease when the lamina is wider (larger $w$), due to higher laminate and larger tilt gaps that effectively reduce reinforcement density.

**Laminate strength and toughness.** The laminate strength can be assessed by considering the conservative first ply failure (FPF) criterion[28] (see details in Supplementary Note 7). We are mainly interested in the effect of tilting on the strength, as demonstrated in the following examples, using the same inputs as in Table 1. In the case of a single in-plane load $N_x$, the normal in-plane strains remain unchanged with tilting, whereas the shear strain increases by 17% (at the BLU top and bottom) up to 40% (at the BLU

midplane). Thus, applying the Hashin-Rotem failure criterion in note 7 on the mid-lamina, we can positively assess that the combination of an unchanged normal strain and an increased shear strain, once referred to the lamina principal material axes, would reduce the overall strength of the laminate due to tilting. Conversely, in the case of a single bending moment $M_x$, the normal in-plane strain in the $x$ direction decreases due to tilting by 0% (at the BLU midplane) up to 16% (at the BLU top and bottom), whereas the shear strain decreases by 26% for all laminae. Thus, applying the Hashin–Rotem failure criterion on the middle, top and bottom laminae, the combination of an unchanged or slightly reduced normal strain and a decreased shear strain would increase the overall strength of the laminate due to tilting. So, in these examples, tilting would improve the laminate strength under a bending moment, while degrading it under a normal in-plane load. Other loading conditions can be assessed in a similar way. Similar analysis, carried out to determine the effect of tilting on the interlaminar shear strength, shows that under a shear load in the $z$-direction the interlaminar strength should improve by tilting.

The BLU fracture toughness has been previously investigated by crack propagation analysis[12,16–19]. However, the effect of tilting and off-axis twisting on the laminate fracture toughness should be assessed by further research. Geometrically, both features increase the potential paths of cracks: (i) laminae become progressively unparallel to the $xy$ plane due to tilting, and thus the interlaminar path of cracks may be increased, and (ii) the in-plane distance between laminae gradually grows as a result of off-axis twisting, so that the laminate is spread over a larger space in the $x$ and $y$ directions as seen in Fig. 5, and thus the cross-laminar path of cracks may be increased. Both crack propagation

mechanisms, delamination and bridging, are expected to dissipate more energy compared to un-tilted, on-axis BLUs. Finally, the effect of the intralayers between neighboring BLUs on the laminate strength and toughness should also be assessed by further research, as the fractures in the samples in Fig. 2 imply that the intralayers are the weakest links in the structure (see more on the fracture type of the cuticle samples in Supplementary Note 8).

## Discussion

The complex helicoidal microstructure of the Bouligand laminate unit (BLU) is examined, revealing a highly asymmetrical conformation. The description is based on high resolution electron microscope study of a representative biological model, the cuticle of the *Scorpio Maurus Palmatus* tarsus, and is clarified with the help of a graphical model. The BLU consists of about 40–100 laminae of chitin-protein fibers embedded in a proteinaceous matrix, possessing two predominant features: off-axis helicity in which the laminae are twisted about their corners rather than centers, and dual-angle laminating where the laminae are tilted in addition to their twisting. The combination of both features results in tightly packed nesting of BLUs, which provides three-dimensional shear interlocking between BLUs. Each BLU is bound by thin fibrous intralayers separating it from its neighbors. The BLUs in a layer are arranged compactly in a 2D array, with thick fibrous interlayers separating it from neighboring layers.

Classical laminate analysis, modified to accommodate the tilting effect, was applied to assess the structure elastic and strength properties. The modification accounts for the varying in-plane stress in a lamina as function of the local upward and downward shifting due to tilting. The laminate stiffness matrices are obtained by integrating the local loads throughout each lamina, over all laminae in the laminate. The calculated in-plane elastic properties include Young's and shear moduli, Poisson's ratios, and shear coupling coefficients, exhibiting approximate in-plane quasi-isotropy limited by the observed total 180° twist angle. By contrast, the calculated flexural stiffnesses (bending and torsion) are far from quasi-isotropy, as the transversal bending stiffness is much higher than the longitudinal. The calculated Young's moduli in the two in-plane directions are in general agreement with previously published nanoindentation measurements of the BLU layers in the scorpion cuticle.

The elastic moduli are degraded due to laminae tilting because tilting effectively reduces the density of the reinforcing material (the fibers) by creating gaps between laminae. By comparison, the flexural stiffnesses significantly improve due to laminae tilting because tilting increases the overall height of the BLU by moving reinforcement material farther away from the midplane. Similarly, tilting improves the BLU strength under flexural loads while degrading it under normal loads. Thus, tilting enables tradeoff between the ability to withstand in-plane loads and the ability to withstand flexural loads. Toughness is also expected to improve due to tilting and off-axis twisting by extending the propagation path of potential cracks. Further research is necessary to solidify this. Another open question is how do tilting and off-axis twisting affect the cuticle's mechanical properties at the higher structural level, which consists of plied layers of BLUs.

Although biological and evolutionary considerations are not within our scope, the question arises why the endocuticle layers consist of BLU arrays rather than wide quasi-continuous unidirectional laminae with similar twist angles. Such arrangement would achieve similar or better in-plane isotropy than the BLU, and would avoid the weak regions of the intralayers. Examples of such continuous laminates exist in nature, and the ability of the scorpion to generate continuous unidirectional laminae is manifested in the fibrous interlayers and in its exocuticle[26]. Reasons that come into mind, which may be subjects for further research, are: (1) laminae tilting would become impossible in such a hypothetical structure, as the upward and downward rotation of a continuous lamina is not geometrically feasible; (2) interlocking and the existence of intralayers between BLUs introduces additional mechanisms for energy absorption and crack propagation, thus enhancing the structural toughness. Questions of similar nature are why the BLU completes just half a turn, considering the partial loss of isotropy compared to a full turn, and why is the bending stiffness in the transversal direction significantly higher than in the longitudinal direction.

The results presented herein are tuned to the biological model selected for this study – the moveable claw of the *Scorpio Maurus Palmatus* – and are supposedly beneficial for its survivability. We anticipate the extension of these observations to other biological structures, as well as to bio-inspired artificial structures. Obviously, synthesizing a structure that would incorporate the unique features of the BLU is very challenging, as off-axis twisting, tilting and nesting seem difficult to implement particularly in microscale, but the potential scientific and engineering gains could be significant.

## Methods

**Scorpions collection and handling.** Adult scorpions belonging to the species *Scorpio Maurus Palmatus* (SP) (Fig. 1a) were collected from the area of Sde Boker in the Negev desert in the south of Israel, in collaboration with the Israeli Parks and Nature Authority and the Hoopoe Yeruham Ornithology and Ecology Center. The scorpions were transported separately in plastic flasks that were sealed inside a hard-plastic box, to prevent escape during conveyance. The scorpions were submerged in liquid nitrogen and then stored in plastic boxes in a freezer at −80 ℃. This euthanizing method is supported in the literature as humane, while not damaging to cuticle structure[29,30]. The Israeli law "Animal suffering (experiments on animals), 1994" excludes invertebrate animals (including scorpions), and therefore no formal ethical approval was required to conduct the experiments. Scorpion chelae were mechanically separated from the rest of the body before characterization. SEM and TEM images were obtained from 3 different animals[26].

**Scanning Electron Microscopy (SEM).** The chela was manually detached from the scorpion body, followed by removal of the tarsus from the chela using a razor blade. The tarsus samples were fixated overnight at 4 ℃ in 2% glutaraldehyde and 3% paraformaldehyde in 0.1 M cacodylate buffer (pH 7.3), followed by immersion of the samples in a fresh fixation buffer for three additional days at 4 ℃. The samples were next rinsed three times for 10 min in 0.1 M cacodylate to eradicate the aldehyde excess. To achieve an improved image contrast in the electron microscope, the tarsus samples were post-fixed in 1% OsO4 in 0.1 M cacodylate buffer for 1 h at room temperature. The unbonded OsO4 was then removed by washing the samples with 0.1 M cacodylate buffer, and the samples were subsequently placed in a critical point drying machine container for dehydration. Graded series of ethanol concentrations of 30%, 50%, 75%, 90%, and 100% were used for the dehydration process, and finally the samples were placed in a critical point dehydration (CPD) machine (Baltech CPD 030) to remove the excess of ethanol from the tissue. The dried samples were manually broken to obtain cross-sections in the longitudinal and transversal directions. The samples were coated prior to SEM imaging with a gold-palladium alloy using an Edwards (Sanborn, NY) S150 sputter coater. High-resolution scanning electron microscopy (HRSEM) images were obtained from the tarsus cross-sections in a SUPRA-55 VP and a Sigma 500 (Zeiss, Oberkochen, Germany) microscopes. Images were obtained at an acceleration voltage of 5 kV and at a working distance of 14–15 mm using a secondary electron (SE2) detector. The ImageJ software was used to measure the cuticle dimensions[26].

**Transmission Electron Microscopy (TEM).** A similar sample preparation procedure as in the SEM was followed. Subsequent to the post-fixation in OsO4, the tarsus samples were rinsed for an hour at room temperature with 2% uranyl acetate (UA). The samples were next dehydrated in graded ethanol series as in the SEM process, and the ethanol excess was removed by rinsing twice with 100% acetone. Then, the samples were embedded in epoxy resin (Epon, Embed 812, EMS, USA), starting with incubation in epoxy in acetone solutions at 30% concentration overnight, followed by 4 h at 50%, and finally overnight at 70%, all at room temperature. Then, the samples were incubated three times in 100% epoxy resin, first for 4 h, then overnight and finally for two additional hours. The embedded samples were placed in a silicon mold and the epoxy crosslinking process was carried out for three days in an oven heated to 60 ℃. Thin sections of ~100 nm were obtained from the embedded tarsus samples using an Ultracut UCT microtome (Leica) and stained with 2% UA and Reynold's lead citrate.

Images were obtained in a FEI Tecnai T12 TEM at 120 kV with a bottom mounted 2k × 2k Eagle CCD camera (FEI, Eindhoven). The ImageJ software was used to measure the cuticle dimensions[26].

**Reporting summary**. Further information on research design is available in the Nature Research Reporting Summary linked to this article.

## Data availability
All relevant data are available from the authors.

## Code availability
All relevant codes are available from the authors.

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

## Acknowledgements
We acknowledge partial support from the G.M.J. Schmidt Minerva Centre of Supra-molecular Architectures at the Weizmann Institute. This research was also made possible in part by the generosity of the Harold Perlman family. H.D.W. is the recipient of the Livio Norzi Professorial Chair in Materials Science.

## Author contributions
I.G. and I.K. contributed equally to this work. I.K. runs the scorpion project as part of his PhD. I.K. captured the scorpions, prepared the chela samples, carried out the electron microscopy, and wrote the methods section. I.G. interpreted the imagery, developed the geometric model, performed the theoretical analysis, and wrote and edited the manuscript. H.D.W. coordinated the study and edited the manuscript. All authors discussed the results and interpretations and reviewed the manuscript.

## Competing interests
The authors declare no competing interests.
