## [Peer Review File · Nature Communications]

Reviewers' Comments:

Reviewer #1:

Remarks to the Author:

This paper introduced a 'different' type of Bouligand structures found in the scorpion (*Scorpio Maurus Palmatus*) claws. In addition to the in-plane twisting laminae (rotated plywood structure), this new Bouligand structure includes an out-of-plane tilting. The authors do a very nice job describing how these 'warped' structures assemble. Classical laminate analytical predictions considering laminae tilting was employed to analyze the stiffness and strength of the composite. It is shown that out-of-plane tilting enhances the flexural stiffness and strength, but decrease the in-plane stiffness and strength. More dissipated energy was expected on this tilting Bouligands structure, due to the additional surface area created by more complex cracks paths. The paper is well written and very interesting. The topic is very timely, and the general audience will appreciate these new structures. However, the paper could be enhanced, if the authors could address the following points directly in the manuscript:

1. Did the authors check the SEM on both surfaces of cut cross-section? I believe it will be important to show that in the paper (or supplementary material). These surfaces can tell more about the architecture and any other residual stress that may be released when they crack open the material.
2. Along those lines, how do the authors rule out that tilting is the result of cutting and breaking fibers. The procedure is not clear. This is a very tightly-packed set of nesting BLUs, how do we know that what we see in the images is not a result of residual stresses stored in the architecture that come that that shape when the material is cut.
3. What is observed in the cross-section of scorpion claws is left-handed BLUs, however the text gets a little bit confusing when it says "Both rotation angles are right-handed rotations" in page 8.
4. The authors say that this architecture generates shear interlocking in all three shear planes, but it is unclear how that could happen? Figure 7c is not clear enough to explain the shear interlocking. The modified classical laminate plate theory (CLPT) only talks about the in-plane elastic properties, so I am not sure how the theory could be relevant here.
5. In-plane elastic modulus is based on shear-lag modified rule of mixture, but this tilting BLUs are discontinuous, however not staggered in-plane. Could the authors provide the in-plane(local) discontinuous fibers distribution to let readers understand the distribution.
6. Scales bars should be included in all the micrographs (e.g., Figure 6).
7. Tilting is realized by θ only, since the in-plane laminate 0 is transversely isotropic in 2-3 plane. However, based on the description, the volume fraction of the fibers in each layer of this BLUs is changing as well. If that is the case, any change in volume fraction may affects the in-plane properties. Accordingly, how much confidence the authors have on the extended CLPT?
8. The authors make claims about the strength and toughness of tilting BLUs. It will be very useful if the authors comment on the reliability of those claims.

Reviewer #2:

Remarks to the Author:

The authors characterize the micro/nano structure and model the mechanical behavior of the cuticle of the Scorpion *Maurus Palmatus* chela. The twisting angles within the ultrastructure give the chela quite more complexity than previously reported Bouligand structures. This work is certainly of interest to the design of biomimetic materials, particularly the design of tough materials.

As the authors euthanized the scorpions, could you please provide ethical documentation to assure that the animals were euthanized humanely?

How can the mechanical role of the shear interlocking be deduced from Fig. 8?

Please provide a better explanation of the white dots in Fig. 3.

Why are the intralayers not included as part of the model?

What is the in vivo biomechanical loading environment for the chela?

What is the composition relative to other Bouligand structures?

Is this Bouligand structure unique to this area of the scorpion's exoskeleton?

Is θ_k equal to ψ_k an assumption? Was this experimentally measured?

How many different chela were analyzed?

Was the OsO₄ used for staining purposes and if so what effect does this have on the microscopy images shown?

Ref: NCOMMS-19-28309

Title: Nested helicoids in biological microstructures

Authors: Israel Greenfeld, Israel Kellersztein, and H. Daniel Wagner

All the issues raised by the reviewers were considered and addressed, and the paper was revised accordingly. Changes in the manuscript and supplement are emphasized in yellow.

Following are our replies to the reviewers and the applied changes.

Reviewer 1

This paper introduced a ‘different’ type of Bouligand structures found in the scorpion (Scorpio Maurus Palmatus) claws. In addition to the in-plane twisting laminae (rotated plywood structure), this new Bouligand structure includes an out-of-plane tilting. The authors do a very nice job describing how these ‘warped’ structures assemble. Classical laminate analytical predictions considering laminae tilting was employed to analyze the stiffness and strength of the composite. It is shown that out-of-plane tilting enhances the flexural stiffness and strength, but decrease the in-plane stiffness and strength. More dissipated energy was expected on this tilting Bouligands structure, due to the additional surface area created by more complex cracks paths. The paper is well written and very interesting. The topic is very timely, and the general audience will appreciate these new structures. However, the paper could be enhanced, if the authors could address the following points directly in the manuscript:

Comment 1.1

Did the authors check the SEM on both surfaces of cut cross-section? I believe it will be important to show that in the paper (or supplementary material). These surfaces can tell more about the architecture and any other residual stress that may be released when they crack open the material.

Reply

We did examine the opposite surface of the cross-sections, but they do not provide additional useful information regarding the reviewer’s comment. However, to address the question of the fracture type, we added the following note and figure in the Supplementary Information and a reference to it in the manuscript:

“8. Fracture type of the cuticle samples. This note expands on the fracture type of the cuticle samples. Refer to the BLU structures in the tarsus cuticle section in the main text including Figure 2. The preparation of the SEM samples was done by manually breaking them following a well-defined and proven protocol (see Methods), rather than cutting them and inducing unwanted deformations. Images were taken from 3 different adult animals, all showing the same typical structure. The question that arises is whether the BLU conformation seen in the SEM images in Figure 2 is the result of a crack propagating internally through the BLU due to residual stresses, or the result of a separation between adjacent BLUs. The answer to this question becomes clearer when observing the view in Figure S3. We see a repeatable pattern of three layers, each with a

sequence of nested BLUs. The BLUs seem intact, without debris that might have indicated internal fractures, and without the erratic paths that are typical of crack propagation. The BLU laminae are seen twisted and tilted, repeatedly in each BLU with the same pattern. There are no signs indicating that either twisting or tilting might have been caused by deformation induced during the breaking of the cuticle. Furthermore, if the BLUs were internally fractured it would mean that they are substantially thicker than viewed, but when observing the longitudinal cross section in Figure 2d, the BLUs are seen to be very slim to the extent that an internal crack propagating along their height is improbable. Put together with the images in Figure 2, we may conclude that the breaking of the cuticle causes separation between BLUs at the intralayer rather than fracture of the BLU itself.

Figure S3. Oblique SEM view of layers of nested BLUs. Zoom-out of the region shown in Figure 2f in the main text.”

We also referred to reference 26 in Methods, where further details on sample preparation are found.

The placement of Figure 2e was corrected (it was flipped horizontally by mistake).

Comment 1.2

Along those lines, how do the authors rule out that tilting is the result of cutting and breaking fibers. The procedure is not clear. This is a very tightly-packed set of nesting BLUs, how do we know that what we see in the images is not a result of residual stresses stored in the architecture that come that that shape when the material is cut.

Reply

Please refer to our response to comment 1.1.

The specimens were not cut but manually broken, as described in Methods, thus avoiding unwanted induced deformations.

Comment 1.3

What is observed in the cross-section of scorpion claws is left-handed BLUs, however the text gets a little bit confusing when it says “Both rotation angles are right-handed rotations” in page 8.

Reply

The sentence and its following sentence were rephrased to clarify the angles definition:

“Both the global (x, y) and local $(1, 2)$ coordinate systems are right-handed, and therefore both rotation angles are defined by the right-hand rule. As observed in the scorpion endocuticle, the helicoid is left-handed and therefore θ decreases with rising z , ranging from 90° at the bottom lamina to -90° at the top lamina, whereas ϕ increases with rising z , ranging from -90° to 90° , respectively.”

Comment 1.4

The authors say that this architecture generates shear interlocking in all three shear planes, but it is unclear how that could happen? Figure 7c is not clear enough to explain the shear interlocking. The modified classical laminate plate theory (CLPT) only talks about the in-plane elastic properties, so I am not sure how the theory could be relevant here.

Reply

To further clarify the interlocking mechanism, the following text and Figure 7f were added:

“Interlocking is achieved by the wavy shape of the interfacial surfaces between BLUs (Figure 7f), such that when an external shear stress is applied, the displacement of BLUs is mechanically precluded. This mechanism is also effective when an external tensile stress is applied, such that the pullout of BLUs is prevented in all directions by the topological obstacles set by their neighbors. Further stiffening of the BLUs array is secured by the inter- and intra-layers whose fibers merge into the BLUs.

Figure 7. (f) Shear interlocking surfaces prevent displacement of BLUs under shear and tensile stresses.”

As to the second part of the comment, indeed, laminate theory refers to in-plane properties, and the relevant shear interlocking is of-course in the xy plane. Nevertheless, the shear interlocking of BLUs in the xz and yz planes is an additional feature of the BLUs packing structure, which may improve the out-of-plane mechanical properties. Although there might be some coupling between the in-plane and out-of-plane interlocking, it is assessed as negligible in view of the slender shape of the BLU. Thus, the in-plane elastic properties obtained by the modified laminate theory are valid.

Comment 1.5

In-plane elastic modulus is based on shear-lag modified rule of mixture, but this tilting BLUs are discontinuous, however not staggered in-plane. Could the authors provide the in-plane(local) discontinuous fibers distribution to let readers understand the distribution.

Reply

Please see Figure 7d,e showing horizontal (in-plane) cross sections through 4 neighboring BLUs, and our reply to comment 2.2. To further clarify, the following sentence was added in the text:

“The fibers are short and discontinuous between BLUs, and are all parallel to the xy plane.”

See also Supplementary note 2 presenting the calculation of the angular spread of fibers in such a horizontal cross-section.

Comment 1.6

Scales bars should be included in all the micrographs (e.g., Figure 6).

Reply

Scale bars were added in Figures 4 and 6.

Comment 1.7

Tilting is realized by z_k only, since the in-plane laminate 0 is transversely isotropic in 2-3 plane. However, based on the description, the volume fraction of the fibers in each layer of this BLUs is changing as well. If that is the case, any change in volume fraction may affect the in-plane properties. Accordingly, how much confidence the authors have on the extended CLPT?

Reply

As described in several places in the text, and seen in Figure 5a, each lamina consists of tight square-packing of fibers embedded in matrix. Therefore, the fiber volume fraction in a lamina is $V_f = \pi/4$, the ratio between the fiber cross-sectional area and the square inscribing it, and is constant as seen in Table 1.

Tilting induces gaps between laminae, and our modeling assumption is that these gaps are filled by matrix to ensure stress transfer between laminae (see the text at the end of the section *Laminate Elastic Modeling*). Calculation of the contribution of the matrix in the gaps to the BLU stiffness showed that it is negligible and unnecessarily complicates the model, and therefore we opted not to include it. For completeness, we added in Supplementary note 5 (after equation (S19)) the following calculation outline for the gaps contribution:

“In case the gaps contribution is desired, the following calculation outline can be used: The matrix in a gap is not reinforced and therefore isotropic. Thus, the stiffness matrix \mathbf{Q}_k in equation (S17) can be replaced by $\mathbf{Q}_{1,2}$ from equation (S12), which does not vary with the twist angle, where we substitute the matrix properties $E_1 = E_2 = E_m$, $G_{12} = G_m$ and $\nu_{12} = \nu_{21} = \nu_m$. A gap has the shape of a wedge with angle δ_ϕ (equation (S1)), and is the same throughout the laminate. Hence, its local thickness is $s\delta_\phi$, varying with the distance s from the z -axis. This varying thickness can replace the constant thickness t in equations (S18) and (S19). Also, the filled gaps slightly degrade the shear-lag length correction factor, η_l , by increasing the ratio R/r in equation (S8) for fibers that are distant from the z -axis. However, after averaging over the whole lamina, this degradation is minor and can be neglected.”

Comment 1.8

The authors make claims about the strength and toughness of tilting BLUs. It will be very useful if the authors comment on the reliability of those claims.

Reply

The reliability of the strength assessment is substantiated by adding the following Hashin-Rotem failure criterion in Supplementary note 7, and a reference to it in the main text:

“To determine whether failure would occur under a combination of stresses, a failure criterion should be applied. For composites reinforced by strong and stiff fibers, such as the BLU, the Hashin-Rotem criterion can be applied, which separates the fiber and interfiber failure modes in a lamina:2

$$\frac{|\sigma_1|}{F_1} = 1$$

$$\left(\frac{\sigma_2}{F_2}\right)^2 + \left(\frac{\tau_6}{F_6}\right)^2 = 1$$
(S37)

where $\sigma_1, \sigma_2, \tau_6$ are the principal stresses in the lamina, and F_1, F_2, F_6 are its corresponding strengths. In the case of the middle lamina and the top and bottom laminae, the laminate principal axes coincide with those of the lamina. This allows laminate strength assessment without the need for angular transformation of the strains. For example, in these specific laminae, an increased shear stress combined with an unchanged normal stress would result in decreasing the strength. Conversely, a decreased shear stress combined with an unchanged or decreased normal stress would result in increasing the strength. Both examples are demonstrated in the main text.”

As to the toughness, we duly noted that the “*laminate fracture toughness should be assessed by further research*”, and suggested potential failure mechanisms.

Reviewer 2

The authors characterize the micro/nano structure and model the mechanical behavior of the cuticle of the Scorpion Maurus Palmatus chela. The twisting angles within the ultrastructure give the chela quite more complexity than previously reported Bouligand structures. This work is certainly of interest to the design of biomimetic materials, particularly the design of tough materials.

Comment 2.1

As the authors euthanized the scorpions, could you please provide ethical documentation to assure that the animals were euthanized humanely?

Reply

Scorpions belong to the phylum of arthropods. According to the Israeli law “*Animal suffering (experiments on animals), 1994*” (<https://www.health.gov.il/LegislationLibrary/Veter02.pdf>), arthropods and other invertebrates are not defined as animals when it comes to scientific experiments on animals. The law is available in Hebrew only, and in the Definitions section in the Introduction the relevant definition is: “In this law – ‘Animal’ is a vertebrate except human”. There is no supplement to this law nor a separate law regarding experiments on invertebrates. Therefore, experimentation on scorpions is allowed. The only legal restriction is that scorpions must not be taken from natural reserves in the country. The scorpions were captured in cooperation and coordination with the Israeli Parks and Nature Authority (note added in the manuscript). Euthanizing was done as described in Methods by submerging the scorpions in liquid nitrogen, a humane method supported in the literature (see details in refs. 26, 29 and 30), while not damaging to the cuticle structure.

As the scorpions are not covered by the Israeli law, formal documentation is not provided by the authorities. Therefore, the Weizmann Institute’s *Institutional Animal Care and Use Committee* (IACUC, <http://www.weizmann.ac.il/vet/iacuc.html>) declined our request for a formal approval. Prof. Tony Futerman, the current head of the IACUC, informed us that “The IACUC cannot

provide this information as the Israeli council for experiments on animals does not define scorpions as animals so we cannot provide such a statement.”

Accordingly, the following sentence was added in Methods:

“This euthanizing method is supported in the literature as humane, while not damaging to the cuticle structure^{29,30}.”

We also added the following statement in the manuscript:

“Experiments on animals

The Israeli law “*Animal suffering (experiments on animals), 1994*” excludes invertebrate animals (including scorpions), and therefore no formal ethical approval was required to conduct the experiments. Further information on the scorpions collection and handling is provided in Methods.”

Comment 2.2

How can the mechanical role of the shear interlocking be deduced from Fig. 8?

Reply

The underlying assumption of the laminate model is that the stresses are effectively transferred from BLU to BLU (see Supplementary note 3). The shear interlocking mechanism is essential for this assumption to apply, and is therefore embedded in the analysis results presented in Figure 8. Removing the shear interlocking effect from the model will invalidate the model. To further clarify the issue, the following paragraph was added in the *Laminate elastic modeling* section:

“Classical laminate theory assumes that the laminate x and y dimensions (lateral width) are much larger than its z dimension (thickness or height)²⁸, a condition that is not met within the scale of a single BLU. However, shear interlocking between BLUs, as well as the thin fibrous intralayers connecting BLUs, ensure that in-plane stresses are effectively transferred between BLUs, making the BLUs layer structurally contiguous. This can be observed in the in-plane cross-sections in Figure 7d,e, in which the short fibers from multiple BLUs form a wide, continuous, nearly unidirectional lamina.”

Comment 2.3

Please provide a better explanation of the white dots in Fig. 3.

Reply

The following text was added after Figure 3:

“Note the white oblong dots in Figure 3b,c, which are slant cross-sections through fibers running inside the cuticle pore canals. The pore canals, also seen in the SEM images in Figure 2c (side view) and Figure 2f (cross-section), are vertical tubular ducts that function as a material transport system^{24,26}.”

Comment 2.4

Why are the intralayers not included as part of the model?

Reply

The presence of intralayers provides one of the mechanisms that enable effective stress transfer between BLUs, and are therefore included in the modeling assumptions – please see our response to Comment 2.2. The following sentence was added in Supplementary note 3:

“Because the interlayers are very thin, their elastic properties do not have a significant effect on the model and were therefore ignored.”

Comment 2.5

What is the in vivo biomechanical loading environment for the chela?

Reply

The following paragraph was added toward the end of the *Laminate elastic properties and isotropy* section:

“In terms of the scorpion biomechanical loading environment, its chela needs to withstand high clamping loads when attacking prey, and at the same time high contact loads when defending against threats. The first type of load is met by the thick tubular structure of the tarsus, which provides a high bending stiffness at the structural level above the cuticle itself. Thus, a high tarsus stiffness can be achieved even with a relatively low cuticle tensile stiffness. The second type of load is met by the bending stiffness of the cuticle itself against contact pressure. Thus, a higher cuticle bending stiffness at the expense of a lower tensile stiffness, as demonstrated by the model, substantiates a reasonable biomechanical tradeoff.”

Comment 2.6

What is the composition relative to other Bouligand structures?

Reply

Please see the paragraph directly after Table 1, beginning with “*Typical material properties are provided in the recent review by Politi et al²⁴, including: ...*”. This reference addresses a wide range of arthropods having a composition similar to that of the scorpion.

Comment 2.7

Is this Bouligand structure unique to this area of the scorpion’s exoskeleton?

Reply

Bouligand structures were also observed during the study in the cuticle of the *Scorpio Maurus Palmatus* tibia (the fixed claw), but were not fully investigated and therefore not referred to in the manuscript. Preliminary observations in the scorpion mesosoma (the main body) did not reveal Bouligand structures.

Comment 2.8

Is θ_k equal to ψ_k an assumption? Was this experimentally measured?

Reply

It is not an assumption but rather an approximation based on the experimental observations, as noted in the *Geometric model* section: “*Because the angular span of both twist and tilt is the same, and as the angular increments are nearly uniform, the observations lead to $\phi_k = -\theta_k$.*”

Comment 2.9

How many different chela were analyzed?

Reply

The following was added in Methods:

“SEM and TEM images were obtained from 3 different animals.”

Comment 2.10

Was the OsO_4 used for staining purposes and if so what effect does this have on the microscopy images shown?

Reply

OsO_4 was used as a staining agent (see methods) for better image contrast and for post-fixation (note added in the manuscript). As to the reviewer’s question, Os has a high atomic number (76), therefore, Os allows a better scattering of the electrons in the microscope and improves electron conductivity. Therefore the contrast of the images is enhanced while reducing sample charging effects.

Reviewers' Comments:

Reviewer #1:

Remarks to the Author:

I have reviewed the author's responses to the my concerns. I think they did a good job at adding the necessary figures and scales bars as requested. The explained the interlocking behavior in various planes, the contribution of the matrix gap, etc. I was also concerned about their comments about toughness, but the authors clarified that it will be subjected to future research. In general, I am pleased with the current version and I will recommend it for publication.

Reviewer #2:

Remarks to the Author:

The manuscript is now acceptable for publication.